

# Chemotype classification and biomarker screening of male *Eucommia ulmoides* Oliv. flower core collections using UPLC-QTOF/MS-based non-targeted metabolomics

Panfeng Liu[1,2,3], Lu Wang[1,2,3], Qingxin Du[1,2,3] and Hongyan Du[1,2,3]

[1] Paulownia Research & Development Center of China, National Forestry and Grassland Administration, Zhengzhou, Henan, China
[2] Key Laboratory of Non-timber Forest Germplasm Enhancement & Utilization of State Forestry and Grassland Administration, Zhengzhou, Henan, China
[3] Non-timber Forestry Research & Development Center, Chinese Academy of Forestry, Zhengzhou, Henan, China

Corresponding author
Hongyan Du, dhy515@caf.ac.cn

## ABSTRACT

**Background:** In the Chinese health care industry, male *Eucommia ulmoides* Oliv. flowers are newly approved as a raw material of functional food. Core collections have been constructed from conserved germplasm resources based on phenotypic traits and molecular markers. However, little is known about these collections' phytochemical properties. This study explored the chemical composition of male *E. ulmoides* flowers, in order to provide guidance in the quality control, sustainable cultivation, and directional breeding of this tree species.

**Methods:** We assessed the male flowers from 22 core collections using ultra-performance liquid chromatography and quadrupole time-of-flight mass spectrometry (UPLC-QTOF/MS) non-targeted metabolomics, and analyzed them using multivariate statistical methods including principal component analysis (PCA), hierarchical cluster analysis (HCA), and orthogonal partial least squares discriminant analysis (OPLS-DA).

**Results:** We annotated a total of 451 and 325 metabolites in ESI+ and ESI− modes, respectively, by aligning the mass fragments of the secondary mass spectra with those in the database. Four chemotypes were well established using the ESI+ metabolomics data. Of the 29 screened biomarkers, 21, 6, 19, and 5 markers corresponded to chemotypes I, II, III, and IV, respectively. More than half of the markers belonged to flavonoid and amino acid derivative classes.

**Conclusion:** Non-targeted metabolomics is a suitable approach to the chemotype classification and biomarker screening of male *E. ulmoides* flower core collections. We first evaluated the metabolite profiles and compositional variations of male *E. ulmoides* flowers in representative core collections before establishing possible chemotypes and significant biomarkers denoting the variations. We used genetic variations to infer the metabolite compositional variations of male *E. ulmoides* flower core collections instead of using the geographical origins of the germplasm

resources. The newly proposed biomarkers sufficiently classified the chemotypes to be applied for germplasm resource evaluation.

# INTRODUCTION

*Eucommia ulmoides* Oliv. is a multifunctional tree species endemic to China. Due to geological processes during the quaternary glaciation, this monotypic genus remained primarily in Mid-eastern China (*Call & Dilcher, 1997*). After its introduction to other regions of China, *E. ulmoides* was planted in 350,000 hm$^2$ of land across 28 provincial administrative districts (*Du, Hu & Yu, 2013*). The bark of *E. ulmoides* has been used as a traditional herbal medicine for almost 2,000 years (*Feng & Liang, 1996*). Currently, the tree is fully utilized as a part of the rubber modification, medicine, health food, pollution-free breeding and landscaping industries (*Du, Hu & Yu, 2013*, *2015*).

*Eucommia ulmoides* is strictly dioecious and its male capitulum inflorescence consists of 5–11 clustered achlamydeous flowers. Several bractlets are attached to its tiny pedicel, and each male flower is composed of 8–12 stamens. In the Yellow River basin and other parts of northern China, the male flower blooming period is between mid-March and early April and lasts approximately 20 days. This is 8–15 days earlier than the bolooming period of the female flower (*Liu, 2010*; *Liang, 2007*). The natural compounds that have been extracted from male flowers, including geniposide, geniposidic acid, aucubin, chlorogenic acid, and flavonoids, have important anti-inflammatory, bacteriostatic, antihypertensive, antihyperlipidemic, hepatoprotective, choleretic, and cardiovascular protective qualities (*Dong et al., 2005*; *Zhang et al., 2007*). In 2014, the National Health Commission of China added male *E. ulmoides* flowers to their novel food list (*National Health Commission, 2014*).

The research and innovation team of the National Germplasm Resources Bank of Major Famous Tree Species in the North, located in Yuanyang, Henan Province, has conserved nearly 2,000 *E. ulmoides* germplasm resources, including more than 750 male collections, since 2009. Previous studies have evaluated the diversity of the morphological traits, amino acid content, and main active components of 193 male collections (*Du et al., 2016*, *2017a*, *2017b*), and 33 male core collections were screened by analyzing the phenotypic and SSR molecular marker data (*Li et al., 2018a*, *2018b*). However, these studies did not explore how metabolic composition is used to classify male flower collections or determine the existence of reasonable chemical indicators when distinguishing traits, which are important to the quality formation and commercialization processes.

The metabolic composition and chemotype of different germplasm resources should be accurately evaluated to effectively expand genetic base, breeding innovation, and direct cultivation. Non-targeted metabolomics is an unbiased approach that comprehensively

excavates the metabolic fingerprints of biological samples, despite having limited linear range and poor repeatability (*Fan et al., 2017*). Because of the particularly high performances in both mass spectrometry (MS) and tandem MS (MS/MS) modes, the combination of ultra-performance liquid chromatography (UPLC) separation and quadrupole time-of-flight mass spectrometry (QTOF-MS) detection has shown absolute superiority in qualitative and quantitative compound analysis (*Taamalli et al., 2014*; *Wolfender et al., 2008*; *Da Silva et al., 2017*). Coupled with multivariate statistical methods, UPLC-QTOF-MS is an optimized compound characterization approach that has been used to analyze ginseng, *Ficus deltoidea*, *Garcinia oblongifolia*, and many traditional Chinese medicinal herbs (*Zhu et al., 2019*; *Afzan et al., 2019*; *Ning et al., 2013*; *Yu et al., 2018*).

In this study, we profiled the metabolite constituents of 22 fresh male *E. ulmoides* core collections flowers grown under similar environmental conditions and cultivation practices. We processed data using multivariate statistical methods, including principal component analysis (PCA), hierarchical cluster analysis (HCA), and orthogonal partial least squares discriminant analysis (OPLS-DA), to determine the possible phytochemical phenotypes and metabolomic markers across the collections. Our study provides a comprehensive view of *E. ulmoides* male flower metabolomes and a basis for the future utilization of germplasm evaluation, nutritional properties, and biological analysis.

## MATERIALS AND METHODS

### Plant materials

We sampled fresh male flowers in the full-bloom stage from 22 representative core collections in the germplasm pool in late March 2018. Collections were sampled from 13 provincial districts and conserved ex situ following grafting propagation in 2013. The male flower morphological characteristics and variations are provided in Table S1 and Fig. S1, respectively. For each collection, we randomly selected six biological replicates from 2–4 individuals, for a total of 132 independent samples for metabolite profiling. After removing bractlets and tiny leaves attached to the inflorescence, all samples were frozen immediately in liquid nitrogen and then stored at −80 °C. Voucher specimens were deposited at the Key Laboratory of Non-timber Forest Germplasm Enhancement & Utilization of State Forestry and Grassland Administration, Zhengzhou. The numbers and descriptions of the voucher specimens are contained in the raw data files (https://zenodo.org/record/3905465#.Xw3Hjud5uUk).

### Extraction procedures

We weighed each sample (ca. 15 mg), manually ground them into powder using liquid nitrogen, and processed all samples in one batch. The powdered samples were macerated with 120 μL of precooled 50% methanol, mixed for 1 min, and incubated at room temperature for 10 min. After removing 20 μL of extracts, we stored the samples overnight at −20 °C. After centrifugation at 4,000 rpm for 20 min, the supernatants were transferred to 96-well plates. We also prepared pooled quality control (QC) samples by combining equal aliquots of all extracts (*Gika et al., 2016*; *Li et al., 2018c*).

## UPLC-QTOF/MS instrumentation and procedures

Chromatographic separations were performed on a UPLC system (SCIEX, Warrington, UK) equipped with an ACQUITY UPLC T3 column (100 mm × 2.1 mm, 1.8 μm, Waters, Borehamwood, UK). The column oven was maintained at a temperature of 35 °C. The mobile phase consisted of water (solvent A) and acetonitrile (solvent B), each containing 0.1% formic acid (v/v). We applied a 10 min gradient elution at a flow rate of 0.40 mL/min as follows: 0–0.5 min, 5% B; 0.5–7 min, 5–100% B; 7–8 min, 100% B; 8–8.1 min, 100–5% B; and 8.1–10 min, 5% B. The injection volume for each sample was four μl. We used a triple TOF 5600 plus (SCIEX, Warrington, UK) to detect metabolites eluted from the column. The Q-TOF was operated separately in both positive (ESI+) and negative (ESI−) electrospray ionization modes. The interface heater temperature was 650 °C with a curtain gas pressure of 30 psi, and ion source gas 1 and ion source gas 2 were both set to 60 psi. The ion spray voltage was 5 kV in the ESI+ mode and 4.5 kV in the ESI− mode. The total cycle time was fixed at 0.56 s. We summed four bins for each scan at a pulser frequency value of 11 kHz by monitoring the 40 GHz multichannel TDC detector with four-anode/channel detection. The dynamic exclusion was set at 4 s. The mass spectrometric data were acquired in IDA mode, and the TOF mass range was between 60 and 1,200 Da. During the acquisition, we calibrated the mass accuracy every 20 samples. To further evaluate the stability of the liquid chromatography-mass spectrometry (LC-MS) during the acquisition, we injected a QC sample after every 10 samples (*Li et al., 2018c*; *Yu et al., 2018*).

## Data pre-processing

We converted the LC-MS original wiff format files to mzXML files using ProteoWizard's msconvert tool (*Adusumilli & Parag, 2017*), and processed them using XCMS, CAMERA, and metaX toolboxes in an R environment. The XCMS and CAMERA analysis matrix included the peak picking, peak grouping, retention time (RT) correction, and second peak grouping, as well as the isotope, adduct, and artifact annotations for each sample (*Smith et al., 2006*; *Kuhl et al., 2012*). The main XCMS parameter settings are provided in Table S2. We identified each ion by combining the retention time and m/z data. The intensity data from each peak were preprocessed using metaX to improve the quantitative information. We removed features that were detected below 50% of QC samples or 80% of the biological samples, and extrapolated values for missing peaks using the *k*-nearest neighbor algorithm (KNN) to further improve data quality. The QC-based robust LOESS signal correction was fitted to the QC data respecting the order of injection to minimize signal intensity drift over time. The relative standard deviations of the metabolic features were calculated across all QC samples, and we removed those were above 30% (*Xiao, Zhou & Ressom, 2012*). The obtained features (m/z at a certain retention time) were queried against the Kyoto Encyclopedia of Genes and Genomes (KEGG, http://www.genome.jp/kegg/) and the Human Metabolome Database (HMDB, http://www.hmdb.ca) (*Kanehisa & Goto, 2000*). The mass errors for the database searches and fragment assignments were always below 10 ppm, indicating the reliability of elemental composition allocation. We used MS-DIAL to compare the MS/MS fragments of

the metabolites against candidate molecules found in the HMDB, Massbank, and an internal fragment spectrum library (*Tsugawa et al., 2015*; *Neumann & Böcker, 2010*).

## Statistical analysis

We applied pareto scaling to the filtered data set, then conducted PCA and HCA using SIMCA-P Version 14.0 (Umetrics, Umeå, Sweden). A default sevenfold leave-one-out cross-validation method was employed to find the optimal model dimensionality and test the significance of the PCA models. We performed HCA clustering to establish the chemotypes of the core collections using the Euclidean distance matrix and the Ward linkage method (*Saccenti et al., 2014*). We used SIMCA-P to construct OPLS-DA models that could evaluate the differences in the established chemotypes. The models was first evaluated using $R^2Y$ (cum) and $Q^2Y$ (cum) values, and then using a permutation test to determine over-fitting. We determined latent biomarkers for each established chemotype using three criteria: (1) a variable importance of projection (VIP) value > 1, (2) a Student's *t*-test *p* value < 0.05, and (3) a fold change (FC) value > 1.5. These were displayed with large, dark-colored marks in volcano plots generated using SIMCA-P (*Triba et al., 2015*; *Ding et al., 2018*).

## RESULTS

### Metabolite annotation

The total ion chromatograms (TICs) acquired in ESI+ and ESI− modes are shown in Figs. S2 and S3. We obtained 11,306 features (including 9,050 high-quality features) and 8,616 features (including 7,487 high-quality features) from the extracted ion chromatograms acquired in ESI+ and ESI− modes, respectively. Since a single isomer feature may match several metabolites from the first-order mass spectra, we calculated the mass feature number corresponding to the candidate metabolites. From the first-order mass spectra, we annotated 6,092 metabolites observed in ESI+ mode, and 3,554 metabolites observed in ESI− mode. From the secondary mass spectra, we annotated 451 and 325 metabolites in ESI+ and ESI− modes, respectively (Table 1). After KEGG pathway enrichment analysis, we found that the metabolites acquired in ESI+ and ESI− modes were involved in 69 and 78 metabolic pathways, respectively. The number of metabolites involved in the top 20 KEGG pathways is displayed in Fig. 1.

### Chemotype classification

Since the metabolites annotated from the MS2 spectra were more accurate and unique than those annotated from the MS1 spectra, we used the MS2 annotation data for chemotype classification of the *E. ulmoides* samples. We followed two matrices: one using 132 samples × 451 metabolites for the ESI+ mode data, and the other using 132 samples × 325 metabolites for the ESI− mode data. The PCA score scatter plot and HCA dendrogram of the ESI+ mode data matrix can be found in Fig. 2. We used HCA to divide the 22 collections into four groups according to their chemotypes. The seven collections AG-1, BJ-5, MC, SNJ-2, SQ-1, SQ-2, and XS comprised chemotype I; the three collections BJ-3, CL, and HZ comprised chemotype II; the six collections BJ-1, BJ-2, BZ,

**Table 1 Statistics of the annotated metabolites.**

| Mode | Number of all features | Number of high quality features | Number of MS1 candidate metabolites | Number of MS1 annotated metabolites | Number of MS2 annotated metabolites |
|---|---|---|---|---|---|
| ESI+ | 11,306 | 9,050 | 9,686 | 6,092 | 451 |
| ESI− | 8,616 | 7,487 | 6,030 | 3,554 | 325 |

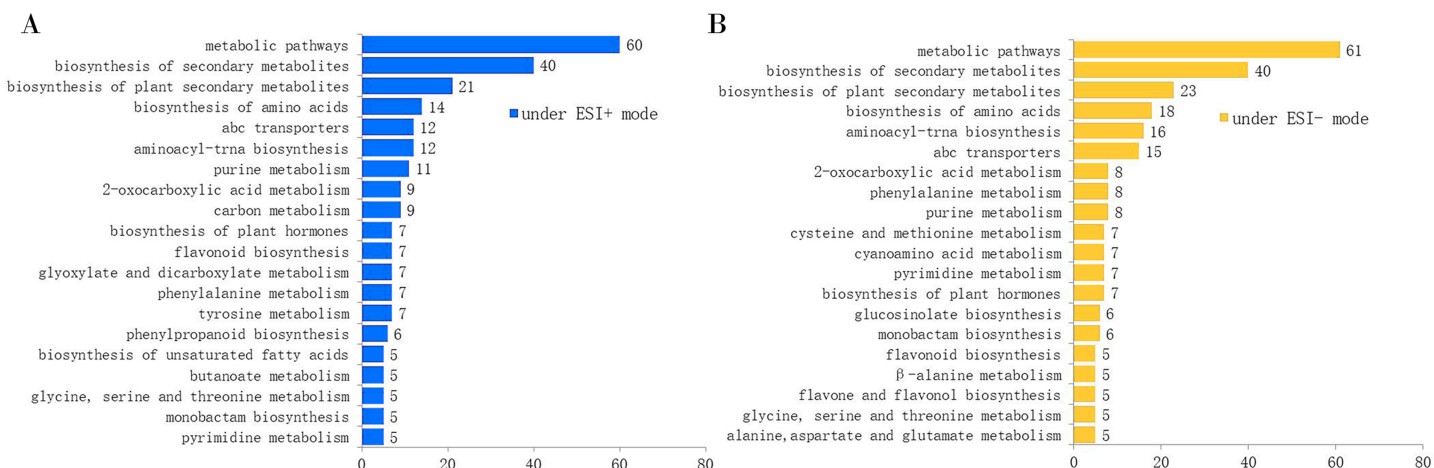

**Figure 1 Number of annotated metabolites involved in the top 20 KEGG pathways by the MS2 spectra.** (A) Under the ESI+ mode; (B) under the ESI− mode.

LC, LY, and SNJ-1 comprised chemotype III; and the six collections AG-2, BJ-4, JA, ZY-1, ZY-2, and ZZ comprised chemotype IV. The total variance of the data using the PCA model was 81.55%, with 26.06% accounted for by principal component 1 (PC1) and 15.82% by principal component 2 (PC2). The two-dimensional PCA score plot provided a visualization of the separation and similarity across the chemical groups. Chemotype I was mainly placed on the left side along PC1, chemotype IV was placed in the center, and chemotype III was placed on the right side. Meanwhile, along PC2, chemotype II was located in the upper half and the other three chemotypes were located in the lower half. Most biological duplications from each collection clustered together, and the boundaries of the classified chemotypes were evident in the PCA score plot, suggesting that the analytical procedures were consistent and repeatable. We did not observe separation based on the geographical origin of the germplasm collections from the analytical results, only that the collections from Shangqiu (SQ-1 and SQ-2) and Zunyi (ZY-1 and ZY-2) clustered in one chemotype.

We applied the PCA model to the ESI− mode data matrix which accounted for 90.40% of the total variance, of which PC1 accounted for 24.36% and PC2 accounted for 16.34% (Fig. 3). Similar to the analysis of the data acquired in ESI+ mode, HCA divided the 22 collections into four chemotypes. Chemotype I was comprised of the five collections AG-1, AG-2, JA, SNJ-2, and ZZ; chemotype II was comprised of the five

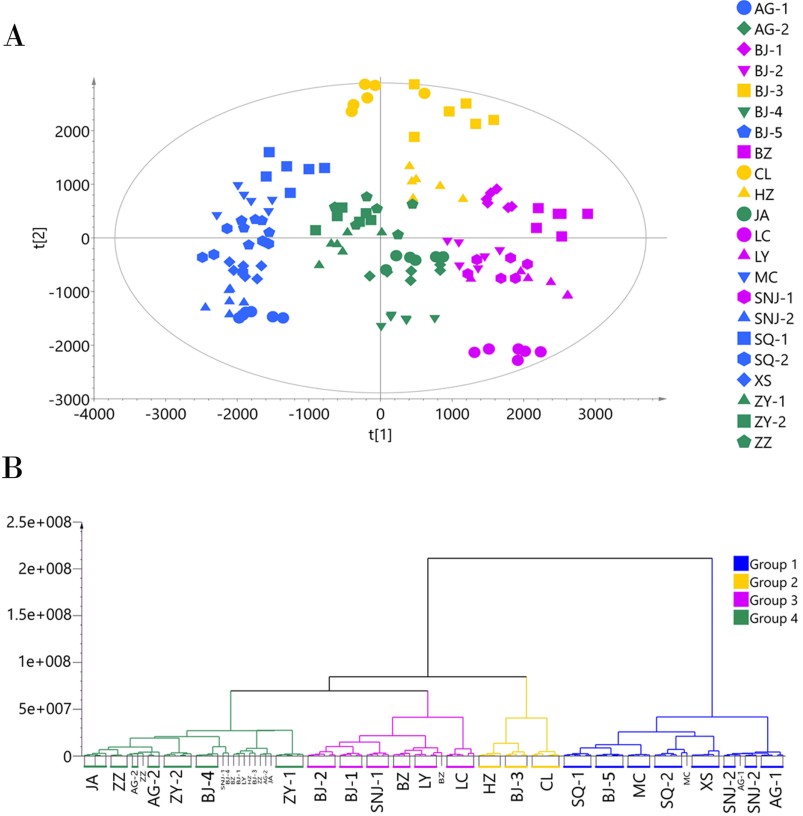

**Figure 2 Chemotype classification of 22 male *E. ulmoides* core collections by PCA and HCA basing on metabolomics data acquired in the ESI+ mode.** (A) PCA score plot of 132 samples, with PC1 accounting for 26.06% of the total variance and PC2 explaining 15.82%.The plot ellipse represents 95% confidence region for Hotelling's T2; (B) HCA dendrogram by using the Euclidean distance matrix and Ward's linkage method.

collections BJ-2, BJ-4, LC, LY, and SNJ-1; chemotype III was comprised of the five collections BJ-5, MC, SQ-1, SQ-2, and XS; and chemotype IV was comprised of the seven collections BJ-1, BJ-3, B2, CL, HZ, ZY-1, and ZY-2. The classification results were slightly inconsistent with those from the ESI+ mode data, especially for chemotype I and chemotype IV. The ESI+ mode identified a higher number of metabolites than ESI– mode,and most metabolites including amino acids, short-chain carboxylic acids and flavonoids were detected in both modes. The chemotype classification boundaries at multiple dimensions in the PCA plots were unclear from the ESI– mode data (Figs. S4 and S5), so we ultimately selected and adopted the classification results originating from the ESI+ mode to establish the chemotype for the collections.

To determine whether a certain classified chemotype of the male *Eucommia* core collections was unique enough to be distinguished from other types, we constructed four supervised OPLS-DA models to provide further insights into the separation (Fig. 4). To assess each OPLS-DA model, we calculated $R^2Y$ (cum) and $Q^2Y$ (cum) coupled with two indicators referring to the intercept of $R^2$ and $Q^2$ from the permutation test (Table 2). We used the generally accepted threshold values (>0.5) of $R^2Y$ (cum) and $Q^2Y$ (cum)

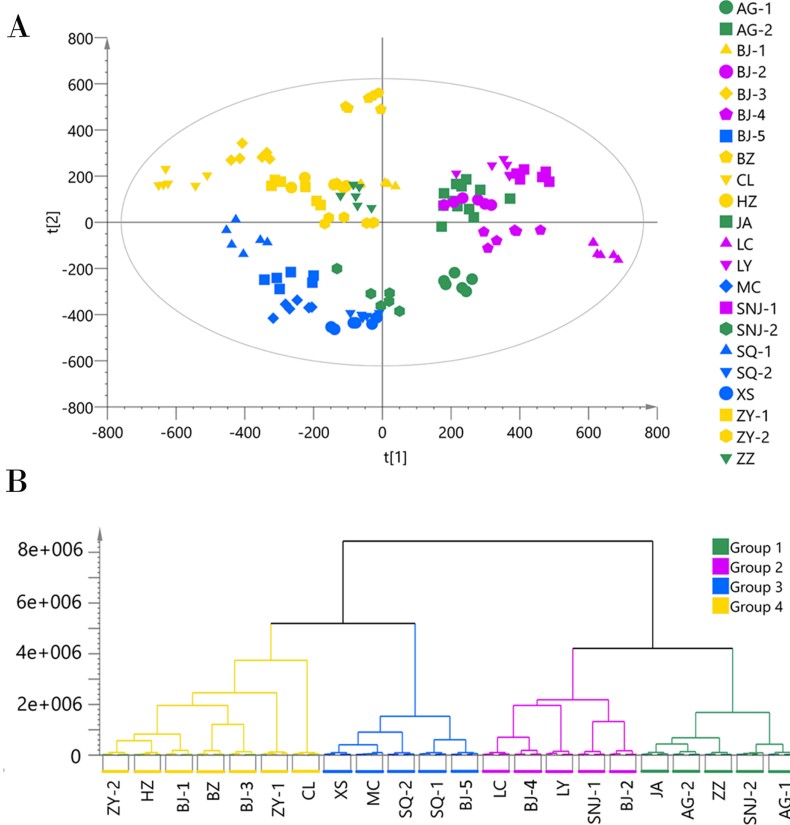

**Figure 3 Chemotype classification of 22 male *E. ulmoides* core collections by PCA and HCA basing on metabolomics data acquired in the ESI– mode.** (A) PCA score plot of 132 samples, with PC1 accounting for 24.36% of the total variance and PC2 explaining 16.34%. The plot ellipse represents the 95% confidence region for Hotelling's T2; (B) HCA dendrogram by using the Euclidean distance matrix and Ward's linkage method.

to judge whether the OPLS-DA model was stable and effective (*Triba et al., 2015*), and clearly found that the constructed models achieved these standards, confirming that each classified chemotype had a distinct metabolite composition. However, the permutation test showed that the OPLS-DA model between chemotype IV and the other chemotypes had slightly higher $R^2$ and $Q^2$ intercept values, suggesting the possibility of over-fitting in that model.

## Screening chemical markers

We determined the most relevant chemical markers for discriminating each chemotype based on the VIP value, *p*-value, and FC value in the OPLS-DA model. In total, we screened 29 chemical markers, 21 of which corresponded to chemotype I, six corresponded to chemotype II, 19 corresponded to chemotype III, and five corresponded to chemotype IV (Fig. 5; Table S3). The chemical markers were flavonoids, amino acids and their derivatives, cinnamic acids and their derivatives, lipids, alkaloids, organooxygen compounds, and organoheterocyclic compounds. The overlapping correlation of the markers is illustrated in Fig. 6. For chemotype I, the exclusive markers

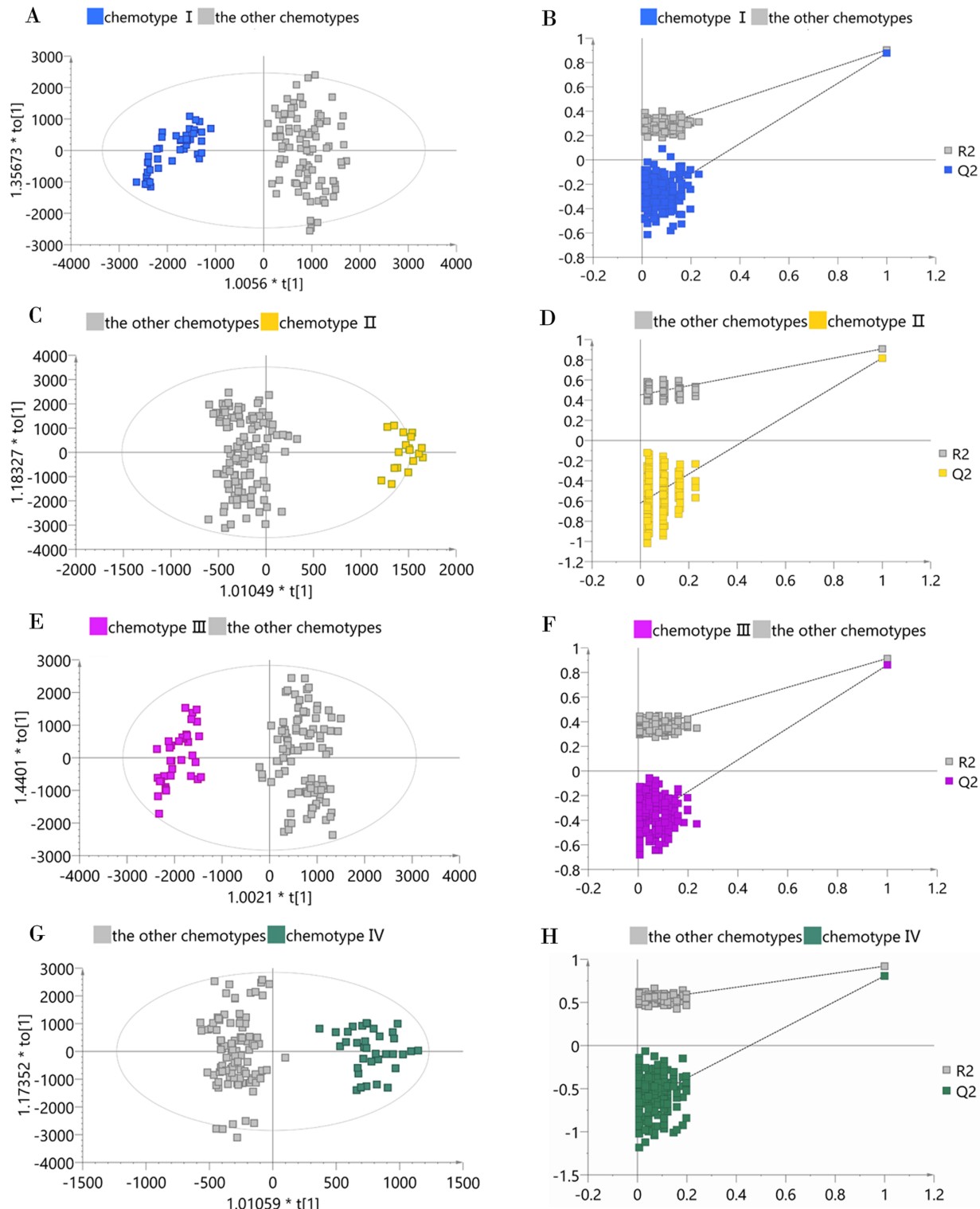

**Figure 4 OPLS-DA score plots and permutation test plots of the classified chemotype against the other types.** (A and B) Chemotype I against the other types; (C and D) chemotype II against the other types; (E and F) chemotype III against the other types; (G and H) chemotype IV against the other types.

**Table 2 Main assessment index of each OPLS-DA model.**

| Model | Orthogonal component | Predictive component | $R^2Y$ (cum) | $Q^2Y$ (cum) | Intercept of $R^2$ | Intercept of $Q^2$ |
|---|---|---|---|---|---|---|
| Chemotype I against the other types | 1 | 2 | 0.904 | 0.878 | 0.231 | −0.361 |
| Chemotype II against the other types | 1 | 4 | 0.908 | 0.818 | 0.451 | −0.619 |
| Chemotype III against the other types | 1 | 3 | 0.914 | 0.861 | 0.326 | −0.424 |
| Chemotype IV against the other types | 1 | 5 | 0.920 | 0.808 | 0.513 | −0.673 |

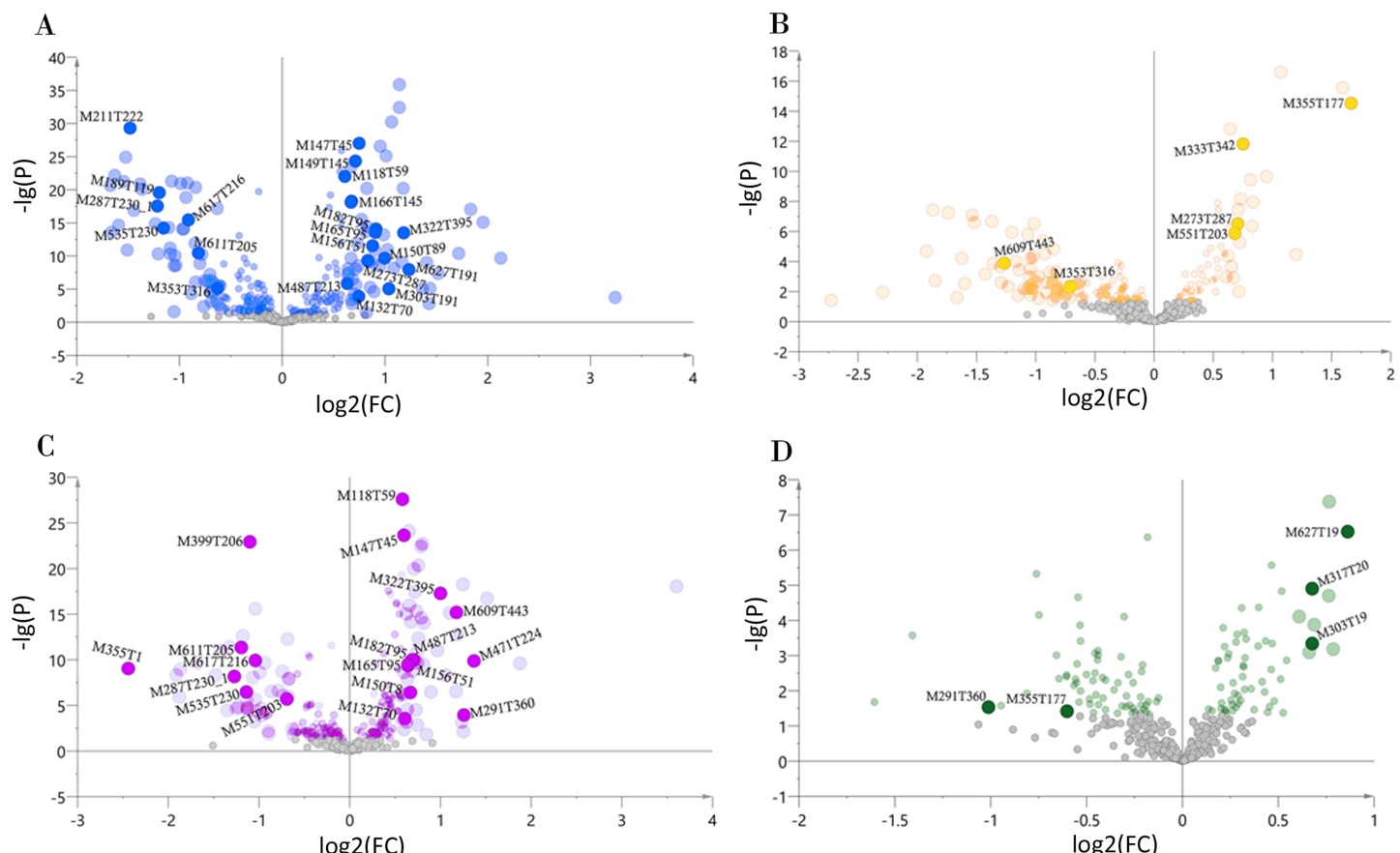

**Figure 5 Volcano plots of chemical markers screened for each classified chemotype.** The X-axis corresponds to log2(FC value), and the *Y*-axis corresponds to −lg(*p* value). (A) Chemotype I; (B) chemotype II; (C) chemotype III; (D) chemotype IV.

were cinnamic acid, L-phenylalanine, glycyl-L-leucine, and jasmonic acid. For chemotype III, the exclusive markers were S-adenosyl-L-methionine and astragalin. Chemotype II and chemotype IV each contained only one exclusive chemical marker, carnosic acid and isorhamnetin, respectively. The chemical marker variations and the peak intensities in the four chemotypes are shown as box-and-whisker plots (Fig. S6). To re-examine the validity of the markers in denoting different chemotypes, we built a data matrix consisting of the 29 markers and 22 collections. We used the same clustering method to draw a heat

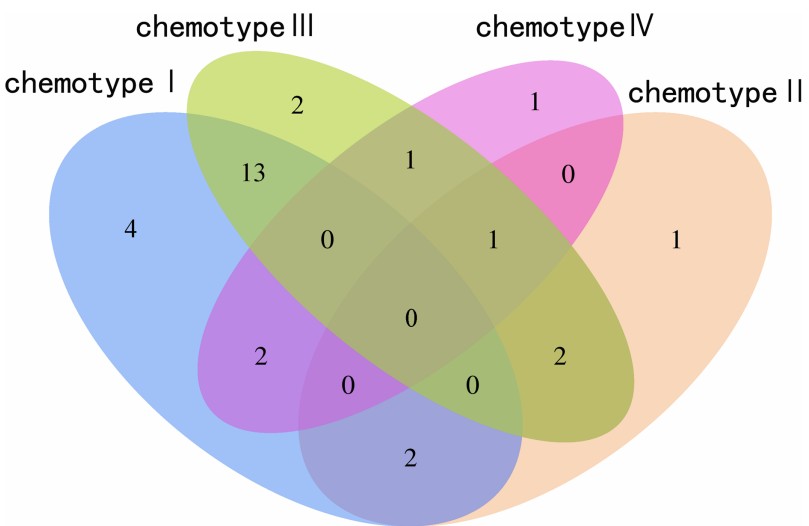

**Figure 6 Venn plot of the screened chemical markers in four classified chemotypes.**

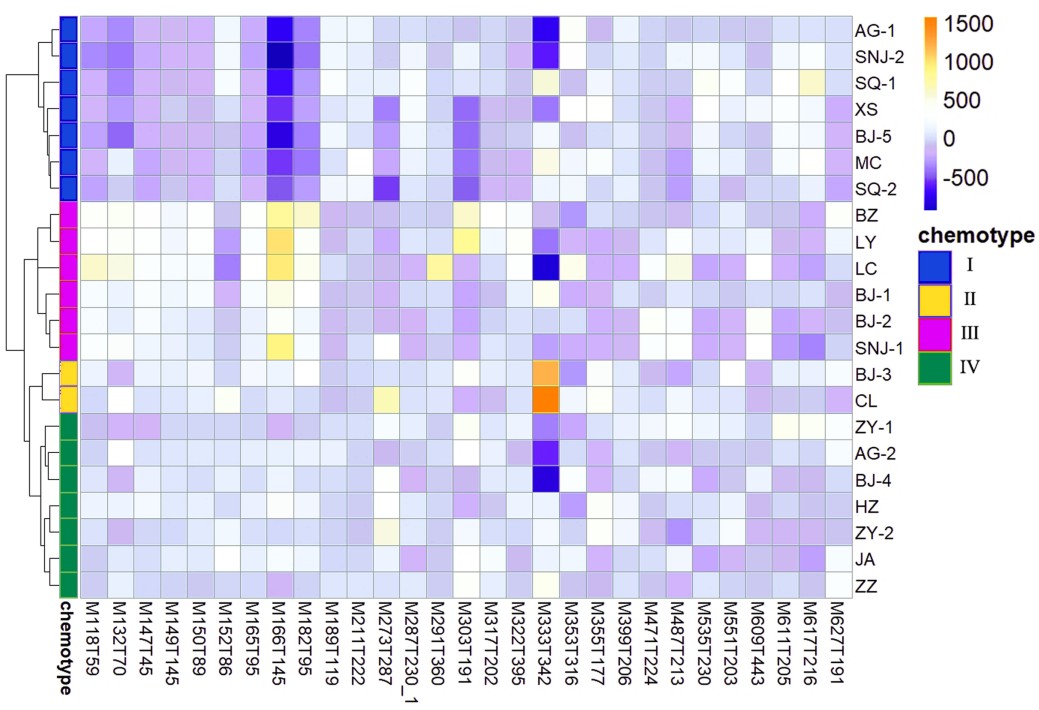

**Figure 7 Heat map visualization for discriminating four chemotypes by 29 screened chemical markers.**

map (Fig. 7) that showed the samples were well-classified and in agreement with the chemotypes established above. However, the HZ collections were misclassified.

## DISCUSSION

*Eucommia ulmoides*, a monotypic genus of *Eucommiaceae*, has long been considered to have relatively limited intraspecific variations compared to other species belonging to

multi-type genera. In the 1980s, there was high demand for *E. ulmoides* bark, significantly increasing its market value. Germplasm resources, especially aged *E. ulmoides* trees, were nearly devastated, leading many researchers to believe that the species' wild resources had been exhausted in China (*Deng, Li & Huang, 2006*; *Wang et al., 2006*). The majority of existing trees were planted from homogeneous provenances and propagated by grafting, which was previously believed to limit genetic diversity (*Wang et al., 2006*). The sustainable cultivation and utilization of the present resources and effective breeding is one of the major concerns of *E. ulmoides* researchers. The first steps should be the implementation and evaluation of the species variations at phenotypic, genotypic, and chemotypic levels. A previous study conducted high-resolution metabolic phenotyping using metabolomic approaches to assess a broad range of possible metabolites and enable informed selection in plants (*Fernie & Schauer, 2009*). Our work was the first attempt at disclosing the metabolome of male *E. ulmoides* flowers and the chemical variations of the core collections from different geographical origins under similar cultivation measures. These results could provide a foundation for further research on the medical use of these male flowers and their metabolic network and regulation.

Due to LC-MS instruments' versatility and sensitivity, they can detect hundreds to thousands of features from a single sample, and, collectively, tens of millions of features for large-scale projects in non-targeted metabolomics studies (*Rocca-Serra et al., 2016*). A significant portion of the detected metabolomes were highly redundant due to the presence of isotopes, adducts, and in-source fragments (*Kachman et al., 2019*). Robust annotation and identification of metabolites in non-targeted studies remains a major challenge in LC-MS-based non-targeted metabolomics, which might lead to false discoveries and incorrect interpretations (*Salek et al., 2013*; *Scheubert et al., 2017*). Other issues, including matching multiple identities with a single feature and multiple disagreeing annotations, often arise when matching the detected features to candidate chemical structures based on their mass and predicting fragmentation patterns based on chemical libraries (*Alden et al., 2017*). However, it is not realistic for most individual laboratories to establish a comprehensive spectral library that encompasses all possible chemical standards (*Broeckling et al., 2016*). Although most modern validation strategies for non-targeted approaches are not well defined (*Naz et al., 2014*), the metabolites annotated in our study cover most well-known chemical constituent categories, provide the metabolomic profiles for male *E. ulmoides* flowers, and present a potential holistic approach to improve germplasm resource evaluation and understand its physiological mechanisms.

The unsupervised clustering method HCA coupled with PCA reduced the dimensionality of the multivariate data and created new principal components using linear combinations to determine similarities and differences across the samples. These methods have been applied in previous plant metabolomic analyses (*Gad et al., 2013*; *Cevallos-Cevallos et al., 2009*). Furthermore, the combined application of PCA and OPLS-DA to spectral datasets yielded valuable insights into general spectral trends and group predictive spectral features (*Worley & Powers, 2016*). The core collections are small sets of accessions chosen to represent the genetic spectrum of the whole collections and

cover high variation levels, representing up to 70% of the genetic diversity of a species. We used multivariate methods to objectively classify the four chemotypes of the male *E. ulmoides* core collections, referencing the main chemical variation forms in the germplasm resource. It should be noted that the biological replicates from the six collections SNJ-1, BZ, BJ-3, BJ-1, HZ, and LY were not clustered within one chemotype. In addition to errors resulting from using the employed clustering method, we assumed that some random sampling errors, injection problems, and other instrument noises or drifts were the main factors affecting the clear classification of the six collections' biological replicates. However, according to our Spearman's correlation analysis based on the peak intensity data, the correlation coefficient was around or above 90%, and we observed a higher correlation in replicates of the same collection than in different collections (Fig. S7). This indicated that there were significant similarities in the collection duplications and they could therefore be identified as belonging to the same chemotype. Due to their different physicochemical properties (e.g., polarity), the two data sets detected using the ESI+ and ESI− modes were also different to some extent, leading to slightly inconsistent HCA chemotype classification. Because of the great dimension and complexity of a non-targeted metabolomics data matrix, it is hard to obtain a strictly precise classification using existing bioinformatics tools. In this study, we verified the effectiveness of the classification results using the OPLA-DA models. However, both classified chemotype sets were inconsistent with geographical origin patterns. We inferred that the chemical variation may have originated from the inherent genetic variation of the germplasm and intrapopulation (not interpopulation) variation. Similar results at the genetic level were found in previous studies (*Wang et al., 2006*; *Wu, 2014*).

Other studies have explored the main categories of the nutritive components of male *E. ulmoides* flowers, including iridoids, phenylpropanoids, flavonoids, amino acid derivatives, and phenolic glycosides (*Yan et al., 2018*; *Ding et al., 2014*). In our current study, we screened out 29 chemical biomarkers belonging to seven metabolite classes, the majority of which were flavonoids and amino acid derivatives. Comparison and variation analyses on the 17 amino acids in *E. ulmoides* male flowers of 193 germplasm resources showed that the content variation degrees, evaluated by coefficient of variation (CV) values, were relatively higher in proline (Pro), cysteine (Cys), lysine (Lys), and methionine (Met) (*Du et al., 2016*). Another study evaluated the content variations of the eight major active components in the same resources, and found that geniposide, geniposidic acid, and astragaline had the highest variation degrees (CV values above 80%); followed by isoquercitrin, chlorogenic acid, and aucubin (CV values around 50%); and total flavonoids and total amino acids had the lowest variation degrees (CV values below 25%) (*Du et al., 2017b*). We assumed that the chemical components with the highest degrees of content variation in the germplasm resources would be potential biomarkers for chemotype differences, and biomarkers such as chlorogenic acid, methionine, and S-adenosyl-L-methionine complied with this assumption. However, the most likely biomarkers geniposide, geniposidic acid, and astragaline did not comply with our results. One possible reason for these inconsistencies may be that the potential biomarkers'

features were not fully annotated because the coverage of standard spectra in the reference database was incomplete. Additionally, these were two studies conducted using different experimental conditions, compound extraction procedures, and measuring instruments. Future studies should validate the identification and structure of the screened biomarkers using multiple optimized in silico tools, and confirm their results using classical targeted analytical approaches such as the NMR technique to get more reliable and reproducible results.

## CONCLUSION

We determined the metabolite profiles of male *E. ulmoides* flowers from different geographical origins using a developed UPLC-QTOF/MS fingerprinting method. We found abundant compositional variations and performed phenotyping at the metabolic level across several male *E. ulmoides* core collections, providing a basis for understanding the formation mechanism of economically important traits. A total of 451 and 325 metabolites were identified in ESI+ and ESI− modes, respectively, from 22 male flower core collections. We coupled the multivariate statistical method PCA with OPLS-DA models to provide a new technique for classifying *E. ulmoides* germplasm resources that is efficient in revealing the major metabolites that contribute to chemotype discrimination. Four chemotypes could be determined from the MS2-annotated metabolites that were detected in ESI+ mode. We proposed that compositional variations across the chemotypes were due to genetic composition variations instead of geographical distributions. We screened a total of 29 robust chemical markers that enabled the differentiation of four established chemotypes, most of which were flavonoids and amino acid derivatives. The newly proposed chemical markers, along with existing monograph and molecular markers, could be utilized for more precise germplasm resource identification. It should be noted that our results are preliminary and require further verification using a larger set of samples.

### Funding

This research was funded by the Fundamental Research Funds for Central Public-interest Scientific Institution (CAFYBB2018QA008). The funders had no role in study design, data collection and analysis, decision to publish, or preparation of the manuscript.

### Grant Disclosures

The following grant information was disclosed by the authors:
Fundamental Research Funds for Central Public-interest Scientific Institution: CAFYBB2018QA008.

### Competing Interests

The authors declare that they have no competing interests.

## Author Contributions

- Panfeng Liu conceived and designed the experiments, performed the experiments, analyzed the data, prepared figures and/or tables, authored or reviewed drafts of the paper, and approved the final draft.
- Lu Wang performed the experiments, prepared figures and/or tables, and approved the final draft.
- Qingxin Du performed the experiments, authored or reviewed drafts of the paper, and approved the final draft.
- Hongyan Du conceived and designed the experiments, authored or reviewed drafts of the paper, and approved the final draft.

## Data Availability

Data is available at Zenodo: Liu Panfeng, Du Hongyan, Wang lu, & Du Qingxin. (2020). UPLC-QTOF/MS non-targeted metabolomic data of 22 male *Eucommia ulmoides* Oliv. flower core collections [Data set]. Zenodo. DOI 10.5281/zenodo.3905465.

## Supplemental Information

Supplemental information for this article can be found online at http://dx.doi.org/10.7717/peerj.9786#supplemental-information.

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
