# Peer review of "Chemotype classification and biomarker screening of male Eucommia ulmoides Oliv. flower core collections using UPLC-QTOF/MS-based non-targeted metabolomics"

_PeerJ, doi:10.7717/peerj.9786_

## Round 0.1 · original submission · Major Revisions

I recommend revising the (putative) identification of metabolites. The raw data of the study should be made publicly available (e.g. on Zenodo.org), to allow the re-use of the data and reproduction of the results.

·

Basic reporting

Line 33: Please make this line clearer - “The compositional variations were inferred originate from genetic differences but not from geographical origins.”

Line 160: “306 and 8,616 features were obtained….” – please correct to 11306

Line 294-295: Why do the authors prefer confirmation of annotated compounds via in silico fragmentation instead of using possibly available reference standards and NMR for novel compounds?



Table S4: Correct spelling of “annotation”. Meta data needs to be explained better. Why do some rows in “Adducts” have a “-“ ?

Experimental design

Result: Metabolite annotation – The procedure of how the annotation of candidate metabolites via MS/MS fragmentation matching was implemented needs to explained more elaborately.

Validity of the findings

no comment

Additional comments

The article is well written and explained. I do appreciate the extensive statistical analysis, however I would like to see more emphasis on the identification of interesting metabolite markers.

·

Basic reporting

It is clear and unambiguous.
Professional English is used throughout.
It conforms to professional standards of courtesy and expression..
Sufficient field background/context is provided.
The article includes sufficient introduction.
Relevant prior literature is appropriately referenced.
Professional article structure, figures, tables.

Figures are relevant to the content of the article,
They have sufficient resolution, and are appropriately described and labeled.

Experimental design

Research question well defined, relevant & meaningful.

Validity of the findings

Not all underlying data have been provided;

Results are robust, statistically sound, & controlled.

Conclusions are well stated, linked to original research question & limited to supporting results.

Additional comments

Mayor points
Not all appropriate raw data have been made available.
Please provide data in table format. (ion intensity matrix)

Please discuss in more detail following metabolites: geniposide, geniposidic acid, aucubin, and chlorogenic acid

Please discuss results from earlier studies (Du et al., 2017; Du et al., 2017; Du et al., 2016).
Which outcomes are different?

L184 It should be noted that biological duplications from the six collections SNJ-1, BZ, BJ-3, BJ-1, HZ, and LY were not clustered within a chemotype.
Please explain this problem with more detail.

L204 The classification results were slightly inconsistent with those from the ESI+ mode data
Please address this problem with more detail. The first two components of a PCA plot is not always the best was to discriminate samples.
The chemotype boundaries in the PCA plot were inexplicit for the ESI- mode data.
For how many of the PCA dimensions? The first two or all?
Did you try random forest classification?


Minor points

L61 is the reference (Du et al., 2017) given twice? Use lettering: Du et al., 2017a; Du et al., 2017b
L63 reference given twice: (Li et al., 2018; Li et al., 2018). Use lettering a and b for all those references across the manuscript.

L150 by using the Elucidean distance matrix -> Euclidean
Were the data previously scaled/normalized? Euclidean distance is strongly influenced by total ion intensity.

L160 306 and 8,616 features were obtained from the extracted ion chromatograms acquired in ESI+ and ESI- modes.
Only 306 features in positive mode? More features were obtained in negative ion mode than in positive ion mode? Maybe you mean 11,306

L240 E. ulmoides bark has been especially expensive in the market.
Please explain in the introduction if the bark or the male flowers are used for medicine purposes.

---

## Round 0.2 · Minor Revisions

The scientific comments have been addressed, but Reviewer 1 still asks you to improve the writing. Please make sure that you polish your manuscript, e.g. by using a professional proofreading service.
I have reviewed the data you uploaded to Zenodo, but I could not find the MS raw files. They should be deposited in the proprietary (e.g. wiff) or community (preferably mzML) file format.

·

Basic reporting

L205-210: Extremely long sentence. Please simplify!
L264-268: Correct typos and grammar
L275: "Contour"? What is it that the authors want to say here?
L308: Correct spelling of "Phenylpropanoids"
L309: "Glycosides" appears twice
L314: The following study..
L323: "inconsistent" is one word
L324 "On" one hand
L335: phenotyping achieved

This time round I had trouble with the English language. It makes it hard to understand the authors' points.
Please go through the text with a fine-toothed comb for grammar and typos. The text is missing prepositions.

Experimental design

Original primary research within Aims and Scope of the journal.

Research question well defined, relevant & meaningful. It is stated how research fills an identified knowledge gap.

Rigorous investigation performed to a high technical & ethical standard.

Methods described with sufficient detail & information to replicate.

Validity of the findings

no comment

Additional comments

In general I think the authors have addressed the reviewer's queries. However, the major impediment remains, which is the language and formulation of sentences. Please pay attention to the language.

---

## Round 0.3 · accepted · Accept

We appreciate the extensive proofreading of your manuscript, which improves the overall impression of your scientific work.